# Self-Direction in Physics Graduate Education: Insights for STEM from David J. Rowe's Career-Long Methods

Carol Nash 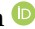

History of Medicine Program, Department of Psychiatry, Temerty Faculty of Medicine, University of Toronto, Toronto, ON M5S 1A1, Canada; carol.nash@utoronto.ca

**Abstract:** The ability to self-direct a research program determines graduate degree completion. Yet, research on incompletion of science, technology, engineering, and mathematics (STEM) graduate programs assumes students' present level of self-direction adequate and neglects to recognize a lack of self-directed learning (SDL) as key. This essay explores SDL for STEM, presenting the work of theoretical nuclear physicist David J. Rowe as a key example of applying a process of SDL in practice. Rowe focused on this challenge of physics graduate education by promoting SDL through the type of research flow that has been found to bring the greatest satisfaction to researchers regarding their insights. Strategies he explored involved his space, time, open mindedness and theoretical contributions with students and in collaboration with colleagues. A self-directed learner himself, Rowe developed methods of mentoring for encouraging physics graduate students to recognize symmetry as valuable in identifying solutions to problems quickly—helping students take the lead in finding insightful resolutions to complex, multidimensional, mathematical physics uncertainties. These strategies for supporting SDL in this context are examined here, with the use of narrative research to interpret the texts and conversations exchanged with the author. The process of SDL developed by Rowe is presented with recommendations on how Rowe's methods may be modeled to improve self-direction in STEM graduate education more widely.

**Keywords:** self-directed learning; STEM; graduate education; David J. Rowe; physics; flow; open mindedness; mentoring; symmetry; narrative research





## 1. Introduction

The ability to self-direct a research program determines completion of a graduate degree [1]. The challenge of encouraging self-directed learning in graduate education is ubiquitous to graduate education in science, technology, engineering, and mathematics (STEM) subjects in their common emphasis of a multidisciplinary approach including inquiry and problem-solving [2]. Recognized as important though underutilized in STEM graduate education, self-directed learning has recently been cited as a "new trend" in STEM subjects that "students appreciate" [3], especially as a result of the increasing importance of virtual learning, requiring independent initiative [4].

Although there is a growing interest in self-directed learning in STEM subjects, rather than considering what is intellectually helpful for promoting the self-direction that leads to successful graduation in STEM subjects, research on incompletion of graduate programs has focused on identifying the sociological factors deemed necessary to change—primarily those to improve self-efficacy [5]—assuming the student's present level of self-direction adequate for success in academe [6]. This has meant that there has been little effort placed on finding ways to improve self-direction in STEM graduate students. Given that a lack of self-direction correlates to non-completion of the graduate program, the type of motivation graduate students have in approaching problem solving can have a profound influence on the developmental careers of those who would choose to become academics [7]. Consequently, finding a way to promote this motivation is critical.

The aim here in investigating self-directed learning with respect to STEM is bipartite. First, the process of self-directed learning Prof. David J. Rowe used throughout his career as a theoretical nuclear physicist will be analyzed. This will be done with respect to both his own research program and to mentor graduate students in an effort to encourage their degree completion. Then, the investigation will indicate how that process might be applied more generally to STEM. This will be accomplished by initially examining self-directed learning in general and then looking specifically at the methods of the type of self-directed learning process supported by Prof. Rowe.

## 2. Self-Directed Learning

A psychological theory that regards the learning process as internally regulated [8,9], self-directed learning (SDL) is based on what the learner personally values, relating to other learners as also self-directing their learning [10]. When this learning takes place in communities based on consensus decision making by finding a time and place to do what each learner values, the self-direction evolves from an individual approach to learning to a way of organizing social interactions when learning [11].

### 2.1. Features of Self-Directed Learners

Self-directed learners are distinguished from other-directed learners by not being tied to a pre-determined schedule [12] and by demonstrating a lack of interest in learning purely for the purpose of gaining extrinsic rewards or praise from those in authority [13]. In contrast, these students learn because of an understanding gained from developing their unique perspective on the world. They determine when learning has been achieved at the level of expertise they have hoped because their learning is tied to what they intend to get out of the learning as it is they who evaluate their learning outcomes [14]. These learners most often prefer to learn from mentors of all ages [15]. They hold no stigma to learning from someone younger or less experienced than themselves as long as that person has something unique to provide that the learner values and desires or needs to understand for their learning to proceed. Similarly, self-directed learners may also search out elderly mentors if they are potentially informative regarding what the self-directed learner values [13] (see Figure 1).

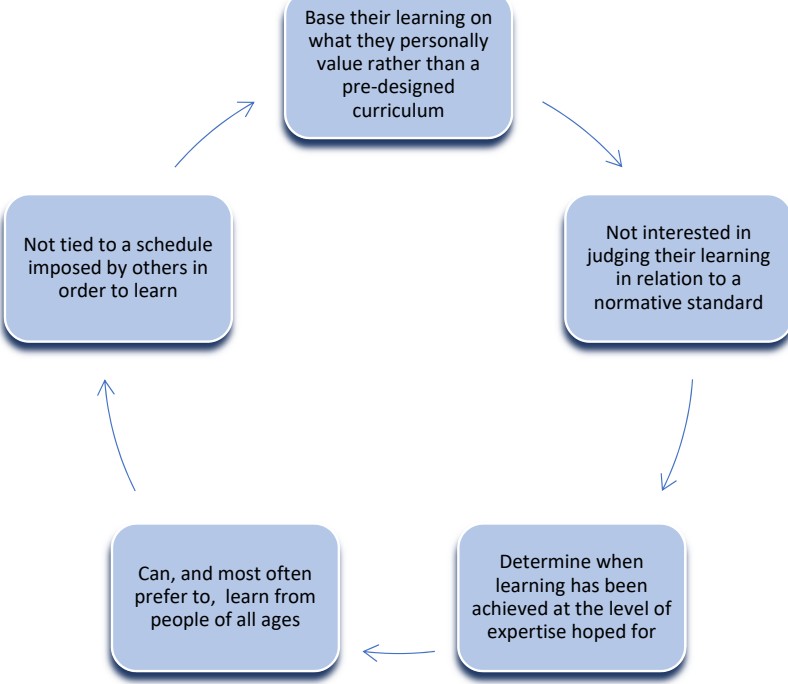

**Figure 1.** What distinguishes a self-directed learner?

To be self-directed learners, STEM graduate students are seen to require awareness of their (1) approaches to learning; (2) suitability for certain strategies for SDL; and (3) satisfaction with SDL [11]. A dissatisfaction with their ability to learn has been identified as being counterproductive in encouraging the positive attitudes and habits of independence required for self-direction in graduate students' learning [16]. Those graduate students who do participate in SDL demonstrate high desire for both learning and self-direction, and in addition they also require self-management skills to be effective at self-direction [17].

SDL requires comprehension of what one values in learning and firmness of purpose to continue to investigate these values in times of boredom, indecision or conflict [18]. It also requires a similar acceptance of others as self-directed learners and acknowledging that the values of others, in providing unique points of view, are necessary to constructing a clear understanding of reality [19]. Together these various points of view are seen to provide our best estimation in constructing reality [20].

### 2.2. Skills for SDL

SDL requires time-management skills, the acceptance of personal responsibility and a way of learning that is very different from supervisor-directed learning [18], which is the other-directed learning that most graduate students experience in depending on their supervisors to define their curriculum, learning pace, research goals and methods of evaluation [21]. Self-efficacy—a person's belief in their potential to organize and execute what is required to attain their intended goal [22]—is necessary to some degree to sustain SDL, but it is not sufficient to provide the interest, skills or self-discipline to self-direct learning. This is especially so since high self-efficacy can distort the student's perception of their actual ability to self-direct their learning [23]. As such, interventions that focus on self-efficacy to improve self-direction in learning are misguided. Graduate students lacking any of the interest, skills or self-discipline to follow self-direction then look to the norm for guidance and evaluation of their behavior, feeling lost and overwhelmed with SDL [24] and are more comfortable with supervisor-directed programs undertaken with a cadre of peers.

With respect to the completion of graduate programs, it is encouraging that the skills for SDL can be learned. A six-step model to develop SDL among students has been identified as useful in gaining the skills for SDL [25].

Develop goals for study
Outline how it will be known those goals have been achieved
Identify the structure and sequence of learning activities
Create a timeline for activities' completion
Identify resources needed to achieve each goal
Locate a mentor to provide feedback on the plan.

Although following these steps is not a requirement for success in SDL, graduate students who feel lost and overwhelmed with SDL might consider this type of organization of their SDL helpful. The reason is the organization relates directly to those qualities that have been recognized as lacking in students who prefer supervisor-directed learning—time-management and accepting personal responsibility.

### 2.3. Assessment of SDL

A scale has been developed to compare the abilities of graduate students regarding SDL [26]. It identifies self-directed learners as exhibiting: initiative, independence, persistence in learning, an acceptance of responsibility for their own learning, a view of problems as challenges rather than obstacles, curiosity, self-discipline, self-confidence, a strong desire to learn, time-management skills, goal orientation, an ability to pace their learning in relation to a plan for completing work, and pleasure from learning. This Self-Directed Learning Readiness Scale (SDLRS) has been tested for validity [27,28]. However, given there is no required or absolute method for success in SDL, response to the SDLRS has been equivocal [29]. This is especially so because learners have been found unreliable in their ability to identify if they possess SDL qualities in practice [30]. More recent research [31] has

indicated, of those subsequent measures that have been developed modifying the original SDLRS, a four-factor 36-item scale has the greatest likelihood of predicting SDL [32]. The factors and their examples are as follows:

Critical self-evaluation

- I evaluate my own performance;
- I like to evaluate what I do.

Learning self-efficacy

- I enjoy learning new information;
- I want to learn new information.

Self-determination

- I prefer to set my own goals;
- I prefer to set my own learning goals.

Effective organization for learning

- I do not manage my time well (reversed);
- I am self-disciplined.

The skills for SDL and the modified SDLRS for assessing SDL are thus available to help any STEM graduate student intending to further their career in academe who may not currently be pursuing their graduate education through SDL. Combined with a moderate amount of self-efficacy, they can sustain their SDL throughout their research careers.

### 2.4. The Role of Mentoring in SDL

Encouraging self-direction in research, mentoring has been identified as unsurpassed in its ability to positively affect graduate students to increase their work satisfaction and fulfilment [33]. Mentoring can be defined as a deep, equitable learning experience with social transformative value [34]. With respect to the competitive research world, graduate students have an inferior position in the research hierarchy [35]. They are isolated in their work and require mentorship in various aspects of academe for successful program completion [36]. Graduate students' perceived experience with academic involvement strongly predicts their educational outcomes [37].

Different types of mentorship have been found valuable. These include opportunities for one-on-one conversations with faculty and cohort socialization activities identifying peer strengths and alliances [38]. Faculty mentorship is perceived by graduate students as important for initiation into the intellectual community where faculty mentorship and research engagement have been found to converge in graduate student success and be necessary for sustainable scholarship [39]. Peer mentorship has been recognized to have wide-ranging effects across four domains of graduate learning: academic, social, psychological, and career [40].

With respect to SDL, mentorship has been found especially effective regarding the creation of student portfolios of their research when the portfolio is integrated by the graduate student into the educational routine and when they are designed to facilitate at least goal-setting, task-analysis, plan implementation, and self-evaluation [41].

### 2.5. Importance of Flow to the Process of SDL

Flow is a desired time when the researcher's mind is stretched to its limits in a voluntary effort to accomplish something valued by them as difficult and worthwhile [42]. In SDL, learners take responsibility for their own learning based on what they personally value. In self-directing, learners may have the ability to experience psychological flow in which the researcher becomes so immersed in the difficult, yet rewarding, work being undertaken that sense of time and place is lost. As such, flow and SDL are not identical. A researcher may be a self-directing learner and still be completely aware of time and place. Flow is not a requirement for the type of research that promotes success as a STEM researcher. It is SDL that is necessary to successfully undertake a research career

in STEM [43]. However, as has been reported by those who have experienced flow in their work, it is flow in conducting this research that brings the greatest happiness to researchers [44].

Not all graduate students have access to the necessary support to achieve flow [19]. This is especially so for non-native English speakers who either lack the knowledge and/or skills to confidently engage in SDL [45]. Furthermore, the type of deep enjoyment of one's work that comes from flow is primarily an intrinsic motivator [46] rather than what researchers personally value of their work as expressed in their SDL. Yet, it was flow that was the experience in SDL that David Rowe hoped to encourage and foster.

## 3. Basis for Investigating Professor David J. Rowe's Process of SDL

Professor David J. Rowe (1936–2020) was one STEM faculty member with a career-long commitment to promoting self-direction in graduate education. A theoretical nuclear physicist with the University of Toronto for over 50 years [47], he focused on the type of motivation that was necessary to encourage graduate students to continue on in academe. A prolific researcher and graduate educator, Rowe considered success as a nuclear and theoretical physicist to depend on physicists' ability to self-direct their learning to develop insightful work regarding physics.

How this would be accomplished in his estimation was primarily based on the application of symmetry to their problem solving in physics [48]. He stressed the value of symmetry in problem solving with his graduate students, his collaborators, and in his own work as a productive theoretical physicist [49] because the use of symmetry through group theory in solving problems in physics permitted a high degree of SDL.

The influential process Rowe developed regarding symmetry in nuclear models will be presented with respect to the relationship it is seen to have to SDL in graduate physics. It will be argued that his methods continue to be useful for success in identifying and solving self-determined STEM problems, aiding in completion of a graduate program. Suggestions will be made for how his process can be advanced within a graduate student program to promote the success in academe that comes from SDL.

In order to present this process of Rowe's in relation to SDL, a basis needs to be provided to understanding this process and how it is being assessed. First will be situating symmetry within science generally and nuclear physics specifically because it was symmetry in nuclear physics which permitted Rowe to believe that self-direction in graduate learning was not only desirable but also entirely possible. Second, an account will be provided of the connection between Rowe and the author of this essay regarding SDL and, third, a description of the methodology used by the author to determine the process Rowe employed for SDL in his graduate students will be presented.

### 3.1. Science, Symmetry and Nuclear Physics
#### 3.1.1. Science

Arranged into theories and facts, science aims to organize and explain phenomena so that its results are reproducible, modifiable or falsifiable by independent observers [50]. This is accomplished through discoveries and inventions, where a discovery identifies the way that things work and inventions indicate how what we perceive can work differently, where sometimes the initial perception of the discovery is itself the invention [51]. Discoveries happen; in contrast, inventions demand that thought be given to whether this new means of perception is of value to those it is intended to serve [52].

#### 3.1.2. Symmetry

As a fundamental aspect of the perception of phenomena [53], symmetry is one such devise that is both a discovery and an invention [54]. During the twentieth century, the development of quantum physics and relativity came with a semantic shift of the word "symmetry" in accordance with the increasing power of it as a conceptual instrument [55]. Symmetry, as a harmony of proportions [56], developed as fundamental to successful

scientific prediction because without this invention all that is identifiable about facts is the binary opposition between "effect present/effect absent". Numbers assigned to experimental results then act only as attributed names rather than fundamentally pointing to a method of ordering [57].

Why symmetries are necessary to understanding experience is a result of consciousness being continuous and immediate in a continuity of time [58]. As such, without a method identifying individual aspects of consciousness and ordering them in some particular way there can be no thoughts about what is perceived and no information gleaned. Symmetry, as a space-time geometry [59], permits the identification of distinct objects to be considered and provides rules for how those objects are ordered in space-time.

The most important concept of symmetry is that space and time are isotropic and homogeneous, that is, all points and all directions in space are equivalent so that there is no real distinction of absolute location in time and space [60]. The symmetry of a system is an observed or intrinsic feature that remains unchanged (invariant) under some type of transformation [61]. It does so by specifying rules of operation, specifically in relation to one, two, three and four dimensions. Symmetry regarding each of these dimensions can be defined as invariant under these conditions: (1) a fixed one dimensional single point represents inversion [62] where the resulting symmetry has either positive or negative parity [63], (2) a fixed single point at one location of the object permitting variability at another location equivalent to $\frac{2\pi}{N}$ results in rotational symmetry [64], (3) symmetry in relation to a fixed two dimensional line promotes reflection [65], (4) variability with respect to a two dimensional line is translational symmetry [66], (5) transformational symmetries are those involving three dimensions [67], and (6) dynamical symmetry, symmetry in four dimension [68]. As that which reflects actual conditions most exactly, broken symmetry [69] is absolute once it was recognized that parity is not conserved in the beta decay of $^{60}$Co (Cobalt-60) [60]. As a result of this discovery, the problem of how to explain the meaning of left and right to those unable to view any one object in common, was solved (for a popular account of this, see the chapter on the fall of parity in Gardner's well-known 1990 book [70]).

Symmetries are divided into discrete—those with only two possibilities (e.g., reflection)—and continuous (rotational, for example) [71]. Restoring broken symmetry is accomplished by considering groups of objects where each relevant symmetry transformation is included as part of the group [72], giving a new perspective on static symmetry where an object is said to have static symmetry if it is invariant under a group of transformations [73]. As such, discrete symmetries are described by finite groups and continuous symmetries by Lie groups [71].

It is upon this basis of understanding symmetries that Rowe developed his research program in nuclear physics and used the results of such to aid graduate students in defining and setting their own research problems in theoretical nuclear physics.

3.1.3. Nuclear Physics

With respect to the structure of nuclei, the processes that go on are dynamical regarding the energies of nuclear states, their spins and parities [74]. Experimentally revealed with $^{168}$Er (Erbium-168) as bands, the relationship of the bands in the energies of nuclear states was discovered to be an axially symmetric rotor [75] and the predictive power of the rotational model of interpretation of the band structure in nuclear structure has been enormously successful [76]. However, it does not reveal what goes on inside rotating nuclei; specifically, whether they rotate as solid objects or like fluids.

Early in his physics career, Rowe was able to demonstrate rigid-body flow with a microscopic rotational theory based on the highly successful adiabatic rotational model [77]. The flows for such a zero-viscosity fluid are irrotational flow, i.e., the currents do not circulate as the fluid rotates. Experiment provides results somewhere between rigid and irrotational flow types, suggesting a fluid flow with non-zero viscosity [78].

With respect to the dynamical group to explain the rotational motions, Rowe discovered that the set of irrotational flows is incomplete, exposing the impediment to constructing

an irrotational-flow model while also indicating what to do about it [79]. Without the use of group theory and the concept of dynamical symmetry, it is unlikely this discovery could have been made [73].

Yet, the full dynamical group for collective motions must contain more than the transformations from one nuclear shape and orientation to another [80]. In addition, transformations giving the nucleus rotational angular momentum and vibrational momentum boosts must be contained. In other words, states of the nucleus in rotational and vibrational motion need to be generated [81]. This Rowe and his graduate student Rosensteel recognized as based on what is known in mathematics as the group of symplectic transformations denoted Sp(3,R) [82].

*3.2. History in Common of SDL Regarding David J. Rowe and the Author*

This author first met Rowe in 1985 when he was Associate Dean, Physical Sciences at the School of Graduate Studies, University of Toronto and this author was a graduate student representative in Education on the School of Graduate Studies Council. That year, they were members of a three-person committee to revise the election procedures for the School of Graduate Studies Council. From that time, it was evident to them that they both shared an interest in the role of SDL in graduate education and as a research process.

As a philosopher of education, this author was seen by Rowe to provide additional insight into his understanding of SDL in advancing his mentoring relationship with his graduate students and research colleagues as well as enhancing his own research path as a mathematical physicist focusing on symmetry in nuclear physics.

In 1990, Rowe was awarded the Isaac Walton Killam Senior Research Fellowship. This fellowship provided Rowe with five years research support to work on the book *Fundamentals of Nuclear Models* [49] with his long-time experimental physics colleague, John L. Wood. With part of the funds he received, he created a research assistantship for this author (Ph.D. 1989) to (1) help him encourage SDL with his graduate students and colleagues, (2) support his own self-direction as a researcher by having this author design a method using Adobe Illustrator for creating the numerous figures for the first chapter the book, and (3) to invite notable physicists to speak at the Department of Physics as Welsh Lecturers [83].

Rowe and this author remained in contact throughout Rowe's life as close family friends. The last telephone conversation this author had with Rowe was a few days before his death when he confirmed he had completed the final paragraph of his planned research program since the time he had developed his views on symmetry as the International Atomic Energy Agency (IAEA) Centre for Theoretical Physics, Trieste Visiting Lecturer from Oct.-Nov. 1966. In his own estimations, Rowe had accomplished all he had set out to do as a self-directed learner in mathematical physics by the time of his death, 8 May 2020. In all, Rowe and this author were friends for thirty-five years. The following information is based in large part on their many years of discussions regarding SDL.

*3.3. Narrative Research as the Method of Investigation of Rowe's Process of SDL*

The author here assumes a distinct method to interpret the texts Rowe left and the years of conversations they had regarding SDL. That method is narrative research. Narrative research is identified as one of the five methods of qualitative inquiry (phenomenological psychology, grounded theory, discourse analysis and intuitive inquiry representing the other four [84]). Defined as the varying perspectives of a story that can be constructed to make experience comprehensible [85] (p. 37), narrative research represents the treatment of data as stories [86] where narrative data result from a communication exchange [87] and an understanding of how human actions are related to the social context in which they occur [88].

It is the particular form of SDL [25] Rowe supported that is attractive to a narrative researcher because it assumes SDL, in addition to being an aim, is also a process to follow in achieving that aim. The process thus represents a story of self-direction accessible to a

narrative researcher. It is for this reason that Rowe's methods are ones that are able to bridge the interest of physicists—and STEM researchers concerned with SDL in general—and that of narrative researchers.

## 4. Rowe's Process of Promoting SDL

How Rowe undertook to promote SDL in his own research and in his supportive interactions with graduate students and in collaboration with colleagues was with respect to four aspects: his space, time, open mindedness and theoretical contributions. Each provided a necessary element to how Rowe promoted SDL. Regarding the six-step model for developing the skills of SDL mentioned in Section 2.2, Rowe's process can be compared by reorganizing the six steps in the following manner.

Identify resources needed to achieve each goal—space
Identify the structure and sequence of learning activities—time
Outline how it will be known those goals have been achieved—time
Create a timeline for activities' completion—time
Locate a mentor to provide feedback on the plan—open mindedness
Develop goals for study—theoretical contributions.

Organized in this manner, it can be recognized that Rowe's process provided for the practical requirements of SDL with respect to space and time as foundational to his own contribution as a mentor. Once the mentorship was established, Rowe would offer the findings of his own research on symmetry for their possible use in making the research work of the graduate student less cumbersome and potentially leading to more insightful outcomes than would otherwise be possible. What space, time, open mindedness and theoretical contributions meant to Rowe in this process are the subjects of the remainder of this section.

### 4.1. Space

On the nuclear physics/high energy floor of the University of Toronto physics department's south side of the building, there were three rooms allotted to Rowe. Most westerly was his personal office, beside that to the east was a room for research associates and directly beside that in the southeast corner was a room for graduate students common to all nuclear physics faculty. The rooms were very different in character. The graduate student space was open, bright and divided into individual areas for each of the graduate students, of which there were about five at any time, two of whom were Rowe's. The next room, for research associates was a dark, utilitarian room that generally served only as a space for them to park their work or to work intently on their own. This lack of ambiance in the research associates' space was primarily because when post-doctoral fellows were part of the research program they spent most of their research time in collaboration with Rowe in his office next door.

Rowe's office was designed to offer the amenities of a faculty club. Important works in physics were visible in both a glass-fronted and two open dark oak bookcases. The three low, side-opening filing cabinets were also of dark oak while an additional dark oak occasional table on the opposite side from the filing cabinets hid another, smaller filing cabinet. The wall-to-wall carpeted room (considered a luxury in those days) held two dark oak arm chairs with upholstered seats on either side of the occasional table, while between these chairs and the window wall was Rowe's dark oak, modern-style desk. The desk created the space for his computer under one of the open bookshelves with his fridge for refreshments located between the computer and the window. On the window ledge was an impressively overgrown Euphorbia Trigona succulent plant. Each of these features of Rowe's office was designed to make Rowe and his collaborators feel as if they were meeting in a faculty club.

The most important aspect of the room's design was its central focus—the large whiteboard attached to the wall above the three, low, side-opening filing cabinets. It was on this whiteboard that the room was arranged. Invariably, it would be covered with the ideas

of Rowe and his colleagues for Rowe to ponder in between meetings and to initiate further discussion at the next meeting. It was the indispensable aid to Rowe's collaborative process.

### 4.2. Time

Of utmost importance to Rowe was his particular view of research time. As a self-directed learner, Rowe was immersed in research time in the manner described as "flow": a desired time when the researcher's mind is stretched to its limits in a voluntary effort to accomplish something valued by them as difficult and worthwhile [42]. When Rowe was working, his mind was invariably in flow. Rowe felt extremely comfortable in flow as this was his natural experience of research. It may be for this reason that, similar to research findings on those who experience flow [89], Rowe's most positive experiences were those related to his research work.

Although Rowe did set up weekly meeting times with his colleague and his graduate students, the need for a specific time was very fluid and could change if Rowe or a colleague or student had a good idea to communicate. Then, Rowe would immediately focus his attention on the graduate student or colleague to hear the idea out—by phone, internet or in person in his office. His excitement about a new initiative stimulated him to readjust his plans in order to understand the new idea. Rowe would preferably invite students and colleagues into his office to have these discussions—among other reasons because it was in his office that he experienced the greatest incidence of flow in his thinking.

Rowe also was ever cognizant of time as something that was scarce and the time-consuming nature of solving physics research problems. He saw his contributions regarding symmetry as ways to decrease the time taken to arrive at profitable results and the ability to use symmetry as the key to taking the lead in finding solutions. It was because of this deep interest that he searched for a way to make the value of group theory evident to graduate students. A result was the co-creation with this author of a little monograph entitled *Symmetry, Art and Nuclear Physics*. Published in house through the Department of Physics, this resource used photos of significant works of art to visualize different types of symmetry [90] and introduce Rowe's findings in relation to group theory.

### 4.3. Open Mindedness

Rowe was not prejudiced regarding from whom he might learn. He believed 50% of physicists should be women, listened attentively to those of different disciplines who might shed light on symmetry, and seemed to have the wonder of a child when hearing of the ideas graduate students and colleagues might bring to a discussion. Although Rowe was internationally acclaimed as a theoretical nuclear physicist, he treated all those with whom he was engaged in a research conversation as an equal. The only criterion Rowe used to evaluate the research discussions he had with others was did they contribute to the discussion in an interesting way. If what was being imparted met this test then Rowe was open and receptive to learning from all others. In this way, it was the ideas discussed that determined Rowe's open mindedness rather than any intention on his part to be fair in principle.

There were four men in particular who were influential in his growth as an open-minded self-directed learner. These included his friend from his undergraduate days at Cambridge, Howard, who taught him to think bravely and deeply on difficult matters; Sir Denys Wilkinson, a British nuclear physicist who encouraged Rowe while at Oxford to consider being a theoretical physicist rather than following through on his original plan to become an experimentalist [91]; Aage Bohr, co-recipient of the 1975 Nobel Prize "for the discovery of the connection between collective motion and particle motion in atomic nuclei and the development of the theory of the structure of the atomic nucleus based on this connection" [92], who convinced him to come to work as a theoretical researcher at the University of Copenhagen after completing his Ph.D. in 1962; and the physicist he admired most, Albert Einstein, whose method of using vivid thought experiments to imagine physical phenomena [93] was one that Rowe himself practiced in making his own

contributions to symmetry using mathematical physics—a method he shared with his graduate students and associates.

*4.4. Theoretical Contributions*

In Rowe's estimation, he never worked harder than from Oct.-Nov. 1966, when he created the series of lectures he delivered as the International Atomic Energy Agency (IAEA) Centre for Theoretical Physics, Trieste Visiting Lecturer. Although he had not undertaken this type of theoretical work before, yet, during that period when he had to deliver new lectures daily, he felt he wrote in a particularly inspired, clear and concise manner. Those lecture notes became, almost verbatim, his first book—*Nuclear Collective Motion: Models and Theory* [80]. This book outlined the research program he would undertake and work to complete until his death.

From the beginning, Rowe's aim was to use the book and the publications that followed to aid graduate students in self-directing their research in a number of different, yet connected, areas: Equations of Motion Formalism; Large Amplitude Collective Motion, Coherent States, Classical & Quantum Mechanics; Microscopic Theory of Nuclear Collective Dynamics; Collective Models as Algebraic Models; Vector Coherent State Theory plus other Mathematical Physics; Shell Model and Coupling Schemes; and Quasi-Dynamical Symmetry and Phase Transitions [94]. Through the use of symmetry employed in each of these areas, Rowe believed that these topics were ones that were inviting and potentially fruitful for graduate students willing to make use of group theory.

4.4.1. Rowe's Account of His Research Program

In assessing the theoretical contributions Rowe worked to provide to the SDL of his graduate students and colleagues, it is important to understand what Rowe considered his fundamental contributions. Rowe's primary goal for his self-directed research program was, in his words, "the development of a microscopic theory of nuclear collective phenomena aimed at understanding nuclear collective dynamics in terms of interacting neutrons and protons" [94]. According to his stated research objectives [94], there were a number of breakthroughs in his research program resulting in publication—each gave graduate students and his colleagues improved theoretical foundations for their own independent research. The first advance was that the many-body theories of equilibrium states and elementary excitations could be expressed in an equations-of-motion formalism. The second success was the theory of large-amplitude collective motion initiated with a graduate student Rick Basserman reducing a complex many-body quantal system to a more conceptually accessible classical system.

This close relationship between classical and quantal mechanics was subsequently explored in collaboration with students and associates (Rosensteel, Ryman, Vassanji, Bartlett), constructing explicit maps from one to the other. In discussions, Rowe always credited his graduate student, George Rosensteel as the one who created the next breakthrough in the formulation of a microscopic theory of nuclear collective dynamics. Starting with a phenomenological model of observables and proceeding to an algebraic expression of the model in terms of a Lie algebra of observables, this strategy yielded a microscopic version of the Bohr-Motttelson collective model known as the symplectic or Sp(3,R) model. Using this model, major developments of this theory were made by Rowe's graduate students and associates (Rosensteel, Carvalho, Vassanji, Rochford and Bahri). Major developments continue to be made by Draayer and his students and colleagues at Louisiana State University using this model [94].

Continuing with his account of his research objectives [94], Rowe indicated that expressing the Bohr model as an algebraic collective model led to a particularly valuable result—a formulation of a Vector Coherent State (VCS) theory. Rowe's graduate students and colleagues (Le Blanc, Hecht, Repka, Turner) used VCS theory to construct representations of various Lie groups, Lie algebras and Lie superalgebras, along with the representations of algebraic models.

Former graduate students and colleagues examined the mathematical structure and applications of dual pairs of group representations intertwining the representations of dynamical groups of symmetry groups in various nuclear physics situations (Rosensteel, de Guise, Repka, Carvalho, Welsh). Furthermore, former graduate students (Turner, de Guise, Rosensteel) along with Rowe separately investigated the shell model coupling schemes associated with each symmetry type and then explored the result when the different symmetries are in competition. It was through these that Rochford, Repka, Bahri, and Rowe discovered the concept of quasi-dynamical symmetry—the mixing of different representations of a dynamical symmetry in a highly coherent manner creating the illusion of symmetry preservation [94].

4.4.2. Rowe's Research Accomplishments after 2009

The account Rowe provided of his research objectives and the graduate students and colleagues who worked in collaboration with him in accomplishing them was written by Rowe in 2009. For his research activities after that time, his long-time collaborator, John Wood, has provided the articles [95] that illuminate the final eleven years of Rowe's research career and those researchers associated with it. These accomplishments are relevant because, although Rowe was a Professor Emeritus at this point in his career, the research he continued to publish was in association with graduate students working under his past students and with his collaborators.

Developments and applications of an algebraic version of Bohr's collective model were undertaken with Rowe in association with colleagues (Welsh, Caprio) [96]. Rowe considered a major goal of nuclear physics as determining the validity of the general form of the shell model as the standard of nuclear structure. He believed that new approaches were needed for deriving effective shell-model spaces. This work was undertaken along with colleagues Carvalho and Repka [97] to the extent that several advances extended the power and versatility of coherent state theory so that it became a vital tool in the representation theory of Lie groups and their Lie algebras. Capelli identities then became a focus of Rowe's research [98], most specifically concerning the construction of holomorphic representations of many Lie algebras by vector coherent state methods.

In the last five years of his research program, Rowe concentrated on improving the mathematical rigor of his work. This was done regarding dual pairs of holomorphic representations of many Lie algebras from a vector coherent state perspective along with mathematician Joe Repka [99] and with respect to microscopic evolution of the collective models and their underlying foundations along with McCoy and Caprio [100].

Rowe's collaboration with Wood was extended to examining isobaric analog states and nuclear shape coexistence [101] following on the publication of *The Fundamentals of Nuclear Models* [49]—summarizing the state of nuclear structure physics and the experimental and mathematical foundations for the models used to understand it a number of years before. The penultimate work of Rowe was a collaboration with Wood and Draayer's team in investigating the key role of the physics of nuclei as an emergent symmetry [102].

Rowe's final work was published one week before his death and was delivered to Rowe's hospital bed by Wood [95]. It was an article concerned with establishing a framework for exploring the dynamics of nuclear rotations [103].

These post-2009 publications by Rowe are considered by Wood to be most compelling with respect to Rowe's influence in the nuclear physics community. According to the past graduate student [104] of Rowe's who invited the author to the 4 June 2022 commemorate symposium in honor of Rowe, Hubert de Guise, Rowe also made important contributions to mathematical physics in working primarily with Joe Repka during this time—stretching back to late 1980s—that remained valuable in this regard to graduate students and colleagues [105–108].

## 5. Examples of the Effect of Rowe's Process Regarding SDL

For Rowe, mentoring was both the foundation to his own work as a self-directed learner and to the style of teaching that he considered most appropriate for supporting SDL in his interactions with graduate students and colleagues. The effects to be summarized will be those related to Rowe's process of supporting SDL in graduate students and with colleagues as a mentor. This will include reference to the most significant effects Rowe's process had on graduate students—those achieved at both Louisiana State University (LSU) with Jerry Draayer's research group and Tulane University, guided by George Rosensteel. These accounts are gleaned from published documents. In addition, two recent email messages to the author from past students of Rowe's—Hubert de Guise and Stijn De Baerdemacker—will directly relate the influence of his mentoring style on their learning as graduate students and on their current relationships with their own graduate students as faculty.

### 5.1. Jerry Draayer—Louisiana State University

According to Jerry Draayer, Roy P. Daniels Professor of Physics and LSU Distinguished Research Master, Rowe played a pivotal role in advancing an understanding of nuclear structure in such a way that these contributions act as an underpinning, paving the way for construction of a bridge to span the gulf between low-energy and high-energy nuclear physics [109].

Draayer considered Rowe one of the few first generation leaders in Fermion-based algebraic models. In his role as a graduate student mentor, four of Rowe's papers were recognized by Draayer as most pedagogically significant [73,79,110,111], spanning from 1985 to 2017.

With respect to the use of symmetry, Draayer identifies Rowe's theory as holding beauty, simplicity and ultimate utility. Commending Rowe as a "a gentleman scholar-scientist" [109], Draayer envisions the legacy Rowe left as a mentor to his graduate students and associates as a campaign to stimulate and engage them. In speaking of the effect that Rowe has had on his own research and graduate students, Draayer comments, "David Rowe was a master mentor and clever innovator—at LSU we are attempting to keep that spirit alive" [109].

An acknowledgement of the verisimilitude of this claim that Draayer's research group is keeping Rowe's spirit alive is the work that has been published by their team based on Rowe's theoretical structure since Rowe's death. Of the numerous publications associated with the research group at LSU indebted to the theoretical work of Rowe, a few of those papers focused on the symplectic model will be cited [112–115]. Rowe's legacy thus remains substantial as a mentor supporting self-directed learning specifically at LSU.

### 5.2. George Rosensteel—Tulane University

Rowe's most illustrious graduate student was George Rosensteel, Professor of Physics at Tulane University. It was Rosensteel who had the breakthrough idea in the formulation of a microscopic theory of nuclear collective dynamics in 1977 [82] that had such a fundamental effect on Rowe's research program. In his tribute to Rowe, Draayer refers to Rosensteel as a friend and major colleague, just an hour's drive away at Tulane University [109]. As such, it is relevant to consider in what way Rosensteel carried forward Rowe's legacy.

The last time Rosensteel referenced the work of Rowe in a publication was one in which Rowe was also a contributing author, published three months before Rowe's death. However, it was also the last time Rosensteel published any research. He himself died on 17 December 2021 at the age of 74 [116].

With respect to Rosensteel's influence on his own graduate students, Rosensteel's obituary noted the following, "He was kind and generous with his knowledge and his time. George brought out the best in people, challenged his students, and encouraged them to reach their potential [116]." In this generosity of time and knowledge, as well as his encouragement of graduate students to pursue what was within their potential, it can be

claimed that Rosensteel had displayed the same values as his mentor, Rowe, with respect to graduate students and this particular attitude to research is now being carried on by the next generation of Tulane's physics graduate students.

### 5.3. Hubert de Guise—Lakehead University

Hubert de Guise, Professor of Physics at Lakehead University, was one of Rowe's graduate students completing his Ph.D. when the author was a research assistant for Rowe in the early 1990s. In an email to the author [104], de Guise offered his views on the mentorship Rowe provided to him.

There are two things that stand out in David's style of supervision.

First, he was unhurried by the outside world ... immensely patient with the de-velopment of his students.

... Second, his skill was in identifying, developing and leveraging the interests of his students, without impeding their progress while still maintaining sufficient focus to actually solve a non-trivial problem.

In summary, de Guise had this general comment regarding Rowe's method of super-vision.

... I do not know of any person other than David who produced so systematically academic offsprings so independent from their supervisor. I'm sure we can find other examples, but this is very rare: clearly David did not seek to produce clones of himself and gave his students sufficient confidence and skills to go beyond the boundaries of their original thesis work.

Regarding the effect of Rowe on his own method of supervision for his graduate students, de Guise offered this assessment.

My students don't work under me (or for me). I do learn a lot from my students and am willing say that I work with them to avoid saying they work under me or for me but obviously this has a different semantic sense than working with David.

Further to these points, de Guise suggested that the relationship he had with Rowe, the one he endorses with his students, is best exemplified by a quotation from philosopher of science Michael Polanyi [117].

... a full initiation into the premises of science can be gained only by the few who possess the gifts for becoming independent scientists, and they usually achieve it only through close personal association with the intimate views and practice of a distinguished master .... A master's daily labours will reveal these to the intelligent student and impart to him also some of the master's personal intuitions by which his work is guided. The way he chooses problems, selects a technique, reacts to new clues and to unforeseen difficulties, discusses other scientists' work, and keeps speculating all the time about a hundred possibilities which are never to materialize, may transmit a reflection at least of his essential visions.

### 5.4. Stijn De Baerdemacker—University of New Brunswick

Canada Research Chair in Theoretical Chemistry, Associate Professor Stijn De Baerdemacker's focus is the translation of ideas from mathematical physics, theoretical physics and machine learning into chemical descriptors that are meaningful [118]. He came to work with Rowe near the end of his Ph.D. in 2007. Presenting on Richardson-Gaudin integrability in quantum many-body systems at the memorial symposium for Rowe on 4 June 2022, De Baerdemacker for the first time met the author and a few days later wrote to the author to provide his views on the importance of Rowe's mentoring to his graduate work and to the current way in which he mentors his own students [119].

David ... showed me his thought processes on multiple occasions during group meetings. I have taken that with me. I think it is important for my students

and coworkers to witness in live action how I build my arguments and come to conclusions.

David indeed liked to build a narrative around the math, and it is also my preferred mode of talking about science (as you might have noticed from my presentation). I also like to share the stories of how things lead up to a certain idea, from a personal and historical perspective. It is important for students to learn that inspiration can come from anywhere, so they need to keep their eyes and sense of wonder open.

. . . There would be a research presentation every week by one of us (or a visitor). I contributed to that on multiple occasions. The magic ingredient was that these were topics that were tangentially related to everybody's interests. I have adopted and copied this approach in my current group. Every week, one of my students (or I myself) will talk about a certain topic that is of interest to the group in general.

. . . About the office space. I did not try to replicate the "faculty club" style in my own offices/labs, because I think it dates back to an old English style which made a lot of sense to David (Cambridge, Oxford, . . . ) but not to these modern days. I want my lab/office to be intellectually stimulating in this "tech-startup-hipster" way.

My approach to mentoring is that (1) my students should feel the thrill of embarking on something new, and (2) they should feel that the project they are working on is "theirs", without internal competition. This means that I want to give them the opportunity to figure out the problem for themselves, at their own pace, and let them own the tools they create for themselves. The flip side of researching a lot of new and fascinating research topics is that I often feel like an overcaffeinated squirrel in a pinball machine, touching a lot of bases but never getting to the real core of something. Then again, every now and then, I can squeeze out a few hours, and share passionately my finding with my students ("Hey, you've got to hear what I've learned from this article!!!"). So, contrary to David, I am not the perfect image of "flow" in the self-directed learning compartment, probably more the "childish enthusiasm" . . . . I guess that's what you mean with the multifacetness of self-directed learning, not necessarily equivalent to flow.

## 6. This Author's Personal Reflections and Experience of Rowe's Methodologies

Although this author was never a graduate student of Rowe's, because she held a research assistantship in association with him for the first half of the 1990s after completing her Ph.D., it is relevant to specify how Rowe's process of SDL influenced the author.

With respect to SDL, Rowe and the author had similar views. This was something they recognized in 1985 during their first committee work together on the School of Graduate Studies Council at the University of Toronto. Both then considered that it is what one personally values as a research problem that should guide a research career, rather than other-directed concerns. For this author, a philosopher of education with a focus on SDL, what attracted her to Rowe and his work was that he was the epitome of the self-directed learner.

When Rowe offered the author the research assistantship in physics it was because both thought that she could bring something to Rowe's research program as a philosopher of education and as a creator of computer illustrations for the first chapter of *Fundamentals of Nuclear Models* [49]. However, in wanting to support Rowe's self-direction in his research program, it was also a concern of the author's that Rowe's ability to engage in flow while conducting research be preserved. It was for this reason that this author volunteered to file all of Rowe's research papers and correspondence—to ensure that Rowe's research flow was not disturbed because of an inability to find some important piece of information.

It was perhaps in his representing the truest example of a self-directed learner that Rowe influenced the author, demonstrating in an exceptional way that SDL is an important

process to engage in as a graduate student and that if a researcher has the appropriate supports, flow can be maintained in this SDL.

## 7. Limitations to Promoting Flow in SDL

The limitations to be presented here are with respect to promoting flow for Prof. Rowe in particular and regarding experimental STEM subjects.

### 7.1. Rowe's Need for Support in Achieving Flow

When Rowe was in flow, he was unaware of anything extraneous to the research problem representing his focus. The implication of this type of SDL is flow requires external support to be continued for any length of time.

When Rowe initially began his career, he had various secretaries to depend on who provided administrative support for his activities. However, later in his career, each professor was expected to take care of their own administration using personal computers. In losing administrative help, Rowe also lost filing support. As a result, surfaces in his office were covered in substantial mounds of paper waiting to be filed. This lack of easy access to his papers was a detriment to Rowe's research program as he often lost that which he most needed. To maintain his flow in self-directing his research program was why this author volunteered to assist in filing the papers in his office, helping him to locate what he needed quickly.

On becoming Professor Emeritus, Rowe shifted his operations from his departmental office to his home office. Rowe was able to remain innovatively insightful in his work during the last part of his life with the dedicated help of his experimental physics collaborator, John L. Wood, and his wife, Una M. Rowe. Without their continuing, effective and sympathetic support of Rowe's need for flow as a self-directed learner, his ability to remain supportive of graduate students and colleagues in their self-direction would not have been possible.

It was Rowe's aim to promote flow with graduate students and associates. Nevertheless, it was a result of the immense and appropriate support that he received by those who worked closely with him throughout his life that this type of flow was possible in his research. Whether they achieved flow or not, from the legacy Rowe achieved with respect to his graduate students and colleagues, it is clear that the mentorship provided by Rowe was of the type to promote the kind of self-direction in graduate programs that encourages completion of degrees as a result of independent research.

### 7.2. Experimental STEM Subjects and Flow in SDL

Rowe's support for SDL was as a theoretician. Faculty who are experimentalists in STEM often need to make significant investments in equipment requiring graduate students to take on specific tasks in the research program. As well, they generally conduct their research in large teams. There have been few studies on self-directed learning in large research teams. However, what has been done indicates that the larger the research group, the less graduate students self-direct their learning leading to frustration on the part of these graduate students as they believe it is difficult for them to fully participate in their learning. What has also been noted is that supervisors of these large teams are generally unaware that their graduate students are frustrated and experience their lack of self-direction negatively [120].

How experimentalists working in large teams can provide the space, time, open mindedness and support to motivate graduate students to follow a process of SDL is not obvious. In this regard, what would be relevant for experimentalists to take note of with respect to Rowe's process would be supporting and providing time for graduate students to investigate the types of projects they personally value that had led them to pursue graduate work.

Rowe began as an experimentalist during his graduate program—it was not until he completed his Ph.D. at the urging of Sir Denys Wilkinson that he became a theoretician. During the years of his graduate program, working under A. B. Clegg at Oxford, Rowe

made sure to set aside time to engage in the experimental self-directed activities he en-joyed. These included constructing radios, taking apart motor vehicle engines and de-veloping photographs. These hands-on activities sustained and promoted his interest in experimental research by allowing him to maintain flow in his research. Faculty in ex-perimental STEM research who are under constraints to promote SDL in the lab, can offer graduate students the time, space, open mindedness and supporting research to the experimental pursuits that graduate students find intellectually invigorating.

Taking account of how Rowe's methods of maintaining flow with his own learning while an experimentalist might be valuable in meeting the challenges associated with SDL for experimental STEM subjects. However, although SDL has been recognized as the form of learning most appropriate for adults [121], a reason that adult learning is seen to be self-directed may be that the research on SDL has been conducted primarily on members of dominant cultures—white, middle-class. Others outside the dominant culture may feel unable to join the workforce without their post-secondary learning being other-directed by educators who are part of the dominant culture [122]. The dependence is often a result of fear that they may not maintain research funding to complete their programs otherwise [123]. It has been found that not all graduate students have access to the necessary support to achieve flow [18]. This is especially so for those non-native English speakers lacking the knowledge and/or skills to confidently engage in SDL [45]. These are important points to consider when promoting SDL and flow in experimental STEM subjects.

## 8. Conclusions

To enhance SDL in STEM to promote degree completion for the purpose of contin-uing an academic career, self-directed learners require awareness of three things, their: approaches to learning; suitability for SDL strategies; and satisfaction with SDL [11]. Dis-satisfaction with their ability to learn is important as it has been identified as diminishing the positive attitudes and habits of independence required for successful SDL in graduate students [16]. For this reason, finding ways to encourage graduate students to participate in SDL by developing their desire for SDL and their self-management skills [17] is key.

Rowe's method of mentoring for SDL in physics is a demonstrated way that self-direction for STEM graduate students in general might be achieved, among other meth-ods [93]. Strict adherence to the six-step model to develop SDL is another approach students have found useful in helping them gain the skills for SDL [25]. Employing the SDLRS that has been developed as a scale to compare the abilities of graduate students regarding SDL [26] may also be useful, as might encouraging the creation of graduate student portfo-lios of their research to facilitate at goal-setting, task-analysis, plan implementation, and self-evaluation [41]. Yet, what is unique about Rowe's method is that it was concerned with the achievement of SDL as well as assuming a particular process in conducting research that was entirely self-directed, encouraging the type of research flow that provides continuing satisfaction with SDL [124].

Rowe was a pioneer, both as a theoretical nuclear physicist and a proponent of SDL, in recognizing the necessary value of SDL for his graduate students and associates. For in-person meetings, Rowe devised a method of encouraging SDL that focused on four aspects: space, time, open mindedness and theoretical contributions. This method produced a heritage of SDL in his graduate students and associates. Regarding virtual meetings, Rowe focused on increasing the theoretical contributions he was able to provide to graduate students and with colleagues in his peer reviewed publications, and his book cowritten with Wood, *Fundamentals of Nuclear Models* [49].

Not every STEM mentor would be able or perhaps willing to provide the type of space Rowe offered his graduate students and associates. However, what was necessarily important to Rowe in providing the space was that those involved felt the space reflected that their contributions were respected and thought important. For Rowe, showing this type of attention towards graduate students' and associates' ideas meant providing a

faculty club atmosphere in the room with a large white board, where ideas were posted and considered for continuing reflection. For those mentoring students in other STEM subjects, creating a different type of space may be for appropriate, as it has been for Rowe's past graduate student, De Baerdemacker, with respect to his own graduate students [119].

Providing undivided attention to graduate students when discussing ideas remained key for Rowe. In determining what should be their own reactions the ideas of graduate students, STEM faculty would do well to consider if they are inspired by these new ideas—similar to Rowe. Those thoughts they find intriguing are ones that they should think of hearing out immediately at the most creatively helpful venue available, rather than waiting for a scheduled meeting.

In examining the contributions of the four men Rowe felt encouraged his own open mindedness (his friend Howard, Sir Denys Wilkinson, Aage Bohr and Albert Einstein), STEM faculty can stress these aspects when supporting the development of interesting ideas in their graduate students through open mindedness: (1) think bravely and deeply about research problems, (2) reflect on experimental data with relevant theoretical models, (3) use these theoretical models to actively predict experimental discoveries, and (4) when making such predictions, use mental imagery to potentially gain new insights into the data and the models. Attending to these guidelines, the graduate students of STEM faculty are more likely to be able to envision the type of interesting contributions that Rowe had been eager to be open and receptive to in learning from others.

Similar to Rowe, STEM faculty can foster SDL in their graduate students by providing students with their own ongoing research to represent tools for decreasing the time it takes—and increasing the range of connections—for graduate students to come to interesting insights. At the same time, doing so exemplifies to graduate students how to construct and maintain an ongoing research program.

The ability to self-direct a research program determines completion of a graduate degree [1] and is a method of learning that students appreciate SDL [3]. This is especially so as SDL is supportive of the satisfaction in research that comes from positive attitudes and habits of independence [16]. Furthermore, mentoring has been identified as unsurpassed in its ability to positively affect graduate students to increase their work satisfaction and fulfilment. In promoting SDL [39], given the importance of SDL to satisfying completion of a research program in graduate education, Rowe's research process and his contributions in group theory can continue to encourage physics graduate students to take up self-direction and encourage researchers in other STEM subjects to incorporate similar methods to facilitate SDL in their areas of research. Learning from Rowe's process of SDL, STEM subjects can promote self-direction in graduate students by providing them with an inspiring research space for meeting (either in-person or online), undivided time for listening to ideas, open-mindedness in mentoring and a fundamental research program for approaching their STEM subject similar to symmetry in its power to promote SDL.

**Funding:** This research received no external funding.

**Acknowledgments:** Thank you to Hubert de Guise for inviting the author to prepare a talk on the legacy of David J. Rowe for the memorial conference in his honor, 4 June 2022 in Toronto which was the stimulus for writing this paper. Thank you also to John L. Wood and Una M. Rowe, David Rowe's most important collaborators, for reading over this work and providing their helpful suggestions. Additionally, thank you to Hubert de Guise and Stijn De Baerdemacker for providing their thoughts on David Rowe's influence regarding their graduate education and its effect on how they currently mentor their own graduate students. Both also provided helpful edits for the revision of this essay.

**Conflicts of Interest:** The author declares no conflict of interest.

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
