# Peer review of "Self-Direction in Physics Graduate Education: Insights for STEM from David J. Rowe’s Career-Long Methods"

_challenges, doi:10.3390/challe13020045_

Round 1

Reviewer 1 Report

This is an interesting take on graduate education.  I appreciate it being written down, as my philosophy is 100% aligned with what is suggested and reported.  I am opposed to micro management of research for both myself as well as my students.  The best students are the ones that are smart and want to define "their path forward" with periodic helps from a mentor being useful.  Some student seem to want to be micro-managed, but I have found that these usually lack the self-confidence to address challenging questions.  The spectrum ranges from the latter to the other end of the spectrum, in students that refuse to acknowledge that they are the center of the universe, when in reality they need training wheels to help point them in the "right" direction.  Though I have not focused in on this myself, my mentoring mindset is clearly aligned with that of David Rowe and I can attest to the fact - from my experience - that long-term success is strongly correlated with the "self-directedness" of a students program.   It is in this regard that understand, appreciate, and see this essay as an attempt to "quantify" as best one can the true value of a "self-directed" approach to learning and discovery.

Author Response

Thank you so much for your positive comments!

I appreciate very much that your views are aligned with those of David Rowe. 

I only wish more STEM faculty felt this way.

Reviewer 2 Report

Thank you for your contribution of this article. Overall, the article reads like a celebration or Rowe, in itself a fine thing, but one that is missing a depth of discussion and consideration of self-directed learning that would make the article more widely applicable. While the narrative nature of the depictions of Rowe’s approach are appreciated, the lack of pulling back the discussion to larger context of self-directed learning leaves little practical application nor consideration for the culture of STEM education. The manuscript, if the intent is to link the description of Rowe’s approach to a successful self-directed learning model, would benefit from data on students experience, even the authors own growth or learning is not described, and/or more detailed and thorough connection to theoretical bases in a more detailed way.

More detailed feedback are as follows:

There is some highlighting in the manuscript, please remove.

In the introduction, please consider providing definitions for both self-efficacy and for self-directed learning to improve comprehension of readers who are not as familiar with the topic and may not understand the nuances of the terms.

2.1.2 Symmetry-If there is a goal to have these concepts reach educators in other STEM fields, consider providing context for lines 116-118, which may not be understandable across all fields without further defining/use of non-technical terms.

3. Rows Process on Promoting Self Directed Learning-There are details of the authors experience with Rowe however, there is little discussion of how the situations described connect to the broader concepts of self-directed learning, how these examples could be applied in other contexts, or connection to student outcomes.

3.1 Space- They way this section is written makes it seem like if a faculty cannot provide amenities in their office or if their office is not big enough to create such a space, they cannot be successful at this approach. Using Rowe’s office as an example then expanding to other applications as well as discussion on how it supported student learning would strengthen this topic.

3.2 Time- The way this section is written as well makes it sounds like if faculty don’t/can’t drop everything on a dime to hear students out they will not be successful at self-directed learning. Discussion the connection of providing students attention and time in other contexts and how it supports positive student outcomes would benefit this topic.

3.3 open mindedness- It seems important to state that the impressions of Rowe are those of the author only. This section also does not relate back to practical application in a way that could inadvertently undermine quality mentorship. Open mindedness is a valuable lens to employ however, as depicted in lines 340-343 is problematic without additional discussion: “The only criterion Rowe used to evaluate the research discussions he had with others was did they contribute to the discussion in an interesting way. If what was being imparted met this test then Rowe was open and receptive to learning from all others.”. My interpretation of this section is that fostering self-directed learning is in part about valuing the ideas of graduate students, however; in the language used the impression is that value is placed only on those who contribute what the faculty considers to be “interesting” contributions. This implies an expectation that students come in with “interesting” contributions already. There is no discussion of how to foster inquisition and critical thinking nor exploration of how the professor’s lens of “interesting” contributions will be shaded by their own biases. This could undermine the inclusion of diverse voices in STEM fields. The implication that having large meetings will inherently lead to self-directed learning without some strategy in fostering inclusive dynamics is lacking in depicting the nuances in academic culture.

344-357- I do appreciate the narrative components that illustrate this manuscript however, this paragraph does not seem to really add anything- there is no connection between the opportunities Rowe was provided and how that contributed to developing open minded communication approaches within his mentoring of graduate students.  

3.4 Theoretical Contributions- The way this section reads is that graduate students will naturally engage in self directed learning based on being introduced to in topic. This could be strengthened by discussion on how mentors can play a role in fostering student interests and engagement in topics, student learning, and community in research groups.

3.4.2. Rowe’s Research Accomplishments After 2009- It is unclear how this section connects to the larger work and connection between Rowe and self0-drencted learning.

4. Effect of Rowe’s Process Regarding Self-Direction in Graduate Students- This section highlights two of Rowe’s graduate students. This could be a great opportunity to bring in data on their experiences under Rowe and connect that to his ability to foster self-directed learning environments.  

5. Importance of Flow to the Process of Self-Directed Learning- This section begins to hint at applying the examples in this manuscript to theoretical bases but is not expanded on. Adding more thorough discussion of the application of the example to theory would greatly improve readers ability to understand the nuances of supporting self-directed learning and to apply the principles described in the examples into their work.

Lines 531-551 do not add to the larger work and even suggest faculty use graduate students or research assistants for “secretarial” supports which may not fit well within the structure or context of many institutions’ expectations of the work graduate students should and should not be responsible for doing for their mentor.

Lines 563-567- There is not data to back up this statement. There is not a clear discussion of mentorship style that Rowe employed, nor data/comments from any additional students to indicate that the examples provided did achieve self-directed learning of students. Only two students in addition to the author are mentioned, and these two students did not provide input on their learning and time with Rowe. As a result, it is difficult to make this statement in argument for the approach described without further data to support it.

Author Response

Thank you for your contribution of this article. Overall, the article reads like a celebration or Rowe, in itself a fine thing, but one that is missing a depth of discussion and consideration of self-directed learning that would make the article more widely applicable. Thank you for making this point. A more thorough discussion of self-directed learning is provided in a new section specifically on Self-Directed Learning. See lines 57-208, including moving the subsections about self-directed learning that had come later in the paper to this new section.

While the narrative nature of the depictions of Rowe’s approach are appreciated, the lack of pulling back the discussion to larger context of self-directed learning leaves little practical application nor consideration for the culture of STEM education. The manuscript, if the intent is to link the description of Rowe’s approach to a successful self-directed learning model, would benefit from data on students’ experience, even the authors own growth or learning is not described, and/or more detailed and thorough connection to theoretical bases in a more detailed way. Thank you for this comment. This paper was submitted for publication prior to the June 4 commemorative conference on David Rowe. As a result of attending that event, during which a synopsis of this paper was presented, two of Rowe’s former students contacted the author and have now provided their accounts of how Rowe was helpful to them with respect to his process of self-directed learning (see lines 898-990). Thank you also for pointing out that the author did not reveal how Rowe helped her develop as a self-directed learner. This has now also been added to the account (see lines 992-1014).

More detailed feedback are as follows:

There is some highlighting in the manuscript, please remove. The highlighting has now been removed.

In the introduction, please consider providing definitions for both self-efficacy and for self-directed learning to improve comprehension of readers who are not as familiar with the topic and may not understand the nuances of the terms. Thank you for this comment. A new subsection has been added regarding Skills for SDL (see lines 105-133). In it, the distinction is made between self-directed learning and self-efficacy (see lines 110-116).

2.1.2 Symmetry-If there is a goal to have these concepts reach educators in other STEM fields, consider providing context for lines 116-118, which may not be understandable across all fields without further defining/use of non-technical terms. A context has been provided in lines 370-373 and a citation (77) added to Martin Gardner’s popular account of symmetry.

  1. Rowe’s Process on Promoting Self Directed Learning-There are details of the authors experience with Rowe however, there is little discussion of how the situations described connect to the broader concepts of self-directed learning, how these examples could be applied in other contexts, or connection to student outcomes. How Rowe’s process relates to the newly added information about the six step model for developing the skill of self-directed learning is formulated in lines 583-599.

3.1 Space- The way this section is written makes it seem like if a faculty cannot provide amenities in their office or if their office is not big enough to create such a space, they cannot be successful at this approach. Using Rowe’s office as an example then expanding to other applications as well as discussion on how it supported student learning would strengthen this topic. What was necessary to the space Rowe provided to support self-direction that can be simulated by faculty in other STEM subjects is noted now, in lines 640-646.

3.2 Time- The way this section is written as well makes it sounds like if faculty don’t/can’t drop everything on a dime to hear students out they will not be successful at self-directed learning. Discussion the connection of providing students attention and time in other contexts and how it supports positive student outcomes would benefit this topic. Thank you for this point. This statement has now been rephrased to, “focus his attention on the graduate student or colleague to”, rather than dropping what he was doing. It has also been made clear why Rowe would be so attentive by adding the statement, “His excitement about a new initiative stimulated him to readjust his plans in order to understand the new idea.” As well, how STEM faculty should respond to their graduate students in similar circumstances has been added (see lines 659-669 for each of these changes).

3.3 open mindedness- It seems important to state that the impressions of Rowe are those of the author only. This section also does not relate back to practical application in a way that could inadvertently undermine quality mentorship. Open mindedness is a valuable lens to employ however, as depicted in lines 340-343 is problematic without additional discussion: “The only criterion Rowe used to evaluate the research discussions he had with others was did they contribute to the discussion in an interesting way. If what was being imparted met this test then Rowe was open and receptive to learning from all others.”. My interpretation of this section is that fostering self-directed learning is in part about valuing the ideas of graduate students, however; in the language used the impression is that value is placed only on those who contribute what the faculty considers to be “interesting” contributions. This implies an expectation that students come in with “interesting” contributions already. There is no discussion of how to foster inquisition and critical thinking nor exploration of how the professor’s lens of “interesting” contributions will be shaded by their own biases. This could undermine the inclusion of diverse voices in STEM fields. The implication that having large meetings will inherently lead to self-directed learning without some strategy in fostering inclusive dynamics is lacking in depicting the nuances in academic culture. Thank you for this point. The following statement has been added to make the limits to Rowe’s open mindedness clear, “In this way, it was the ideas discussed that determined Rowe’s open mindedness rather than any intention on his part to be fair in principle.” (see lines 693-695). Yet, the reviewer is correct that this implies those graduate students without interesting ideas are left out of the discussion. To indicate how to foster interesting ideas in STEM graduate students, a new paragraph—related to what Rowe internalized from the four men who inspired him in his open mindedness—has been added (see lines 710-718).

344-357- I do appreciate the narrative components that illustrate this manuscript however, this paragraph does not seem to really add anything- there is no connection between the opportunities Rowe was provided and how that contributed to developing open minded communication approaches within his mentoring of graduate students.  See the previous point made above for how the new paragraph, from lines 710-718, provides the four aspects that STEM faculty can use in supporting self-directed learning.

3.4 Theoretical Contributions- The way this section reads is that graduate students will naturally engage in self directed learning based on being introduced to in topic. This could be strengthened by discussion on how mentors can play a role in fostering student interests and engagement in topics, student learning, and community in research groups. A new paragraph has been added indicating how STEM faculty can use their own ongoing research programs to provide tools to their graduate students and, at the same time, act as an example to graduate students for how a research program is constructed and maintained (see lines 739-743).

3.4.2. Rowe’s Research Accomplishments After 2009- It is unclear how this section connects to the larger work and connection between Rowe and self0-drencted learning. The reason why Rowe’s accomplishments after 2009 are relevant is mentioned now in lines 796-799.

  1. Effect of Rowe’s Process Regarding Self-Direction in Graduate Students- This section highlights two of Rowe’s graduate students. This could be a great opportunity to bring in data on their experiences under Rowe and connect that to his ability to foster self-directed learning environments. As a result of the June 4th commemorative symposium for David Rowe, the author was able to meet former students of Rowe’s. Two of these have provided the author with accounts of how Rowe’s mentoring process affected them as graduate students and how this has influenced their mentoring of their own graduate students. See sections 5.3. and 5.4.  
  2. Importance of Flow to the Process of Self-Directed Learning- This section begins to hint at applying the examples in this manuscript to theoretical bases but is not expanded on. Adding more thorough discussion of the application of the example to theory would greatly improve readers ability to understand the nuances of supporting self-directed learning and to apply the principles described in the examples into their work. Thank you for this point. A way that Rowe’s process can be applied for experimental STEM faculty is suggested now in relation to how it can be modified to encourage the self-direction of graduate students based on Rowe’s way of continuing with his self-direction when he was an experimental graduate student himself. See lines 1125-1192.

Lines 531-551 do not add to the larger work and even suggest faculty use graduate students or research assistants for “secretarial” supports which may not fit well within the structure or context of many institutions’ expectations of the work graduate students should and should not be responsible for doing for their mentor. Thank you for pointing out that this information unintentionally makes it seem as if support is being given for making graduate students into secretaries. Lines 1087-1094 now explain that the author was not a graduate student in performing these supports and that graduate students should never be expected to provide this role. Why the author did this as a researcher interest in flow in self-directed learning is now specified.

Lines 563-567- There is not data to back up this statement. There is not a clear discussion of mentorship style that Rowe employed, nor data/comments from any additional students to indicate that the examples provided did achieve self-directed learning of students. Only two students in addition to the author are mentioned, and these two students did not provide input on their learning and time with Rowe. As a result, it is difficult to make this statement in argument for the approach described without further data to support it. Thank you for this point. There are now three new subsections including two other graduate students of Rowe’s and my own (see lines 900-1015).

Thank you to the reviewer for providing this detailed assessment. Each of the suggestions provided has truly improved the essay.

Reviewer 3 Report

I have read this paper many times and am baffled by the organization. One example is the abstract.  It starts very abstractly.  Only late in the paragraph does it even mention Prof. Rowe.  Typically, the abstract gives the organization of the paper.  I find an assortment of disjointed topics—another one tries to discuss graduate education in general.  The type of education that it discusses is only for a theorist.  As an experimentalist, I am very interested in learning the details of a theorist’s education.  I do not see enough points to obtain that information.  An experimental graduate student must be assigned specific tasks as a significant investment in equipment must be received for their research.  The type of graduate education described is only for a theoretical education.  If it is intended for an experimentalist, then there must be a description of how it can be adapted.

The individual sections of the paper are well written.  However, they are not assembled into a coherent narrative.  I see the article continually changing the perspective.  I do not know if this paper honors Prof. Rowe or describes his innovative teaching methods.  I think the report needs clear organization before publishing.  This article discussed Prof Rowe’s research program and self-directed research method.  It is not clear which is the focal point of the paper.

In addition, the description of Self-Directed Learning needs more explanation.  There should be specific examples of this philosophy of education that differs from the perspective of a particular graduate student.  The paper would be much stronger if there were a detailed discussion on the traditional way to mentor a student compared with Self-Directed Learning.  Several such examples would make the paper much more substantial.

Author Response

I have read this paper many times and am baffled by the organization. Thank you for your time and interest in reading the paper many times. I hope that with the changes that have been made, the reviewer is no longer baffled by the organization.

One example is the abstract.  It starts very abstractly.  Only late in the paragraph does it even mention Prof. Rowe.  Typically, the abstract gives the organization of the paper. Thank you for stressing that the abstract should give the organization of the paper. The title of the paper has been reorganized so that it is clear the concern is primarily self-directed learning in STEM with Prof. Rowe’s career-long commitment to a particular process for physics graduate education an example of self-directed learning in STEM. With this rearrangement of the words in the title, it now makes sense that Prof. Rowe is not mentioned until mid-way in the Abstract (see lines 2-3)   

I find an assortment of disjointed topics—another one tries to discuss graduate education in general.  Thank you for stating you find the topics disjointed. A new section, section 2, now provides all the information about self-directed learning. It is no longer broken up into an initial discussion and, later, as it pertains to David Rowe (see lines 57-208).

The type of education that it discusses is only for a theorist.  As an experimentalist, I am very interested in learning the details of a theorist’s education.  I do not see enough points to obtain that information.  An experimental graduate student must be assigned specific tasks as a significant investment in equipment must be received for their research.  The type of graduate education described is only for a theoretical education.  If it is intended for an experimentalist, then there must be a description of how it can be adapted. Thank you for pointing out that I have not acknowledged that the ways experimentalists work is distinct from theoreticians. As a result, I have added new information under section 6.2. that specifically pertains to experimentalists and how they might encourage self-directed learning under the constraints that they face in conducting research (see lines 1125-1194).

The individual sections of the paper are well written.  However, they are not assembled into a coherent narrative.  I see the article continually changing the perspective. Thank you for letting me know that you don’t see the paper as presenting a coherent narrative and that you see the article continually changing perspective. I believe that, by moving all the sections about self-directed learning to a new section that comes directly after the Introduction, this problem has been addressed (see lines 57-208). 

I do not know if this paper honors Prof. Rowe or describes his innovative teaching methods.  The paper is intended to discuss the importance of self-directed learning to graduate education in STEM and that Prof. Rowe was one faculty member who was particularly committed to self-directed learning. In seeing self-directed learning as a process, not just an aim, of graduate education, Rowe’s methods are of interest as he had a deep and conscious concern to support self-directed learning. It is hoped that, with the new title and with the creation of a new section devoted to self-directed learning immediately following the Introduction, this problem has now been resolved.

I think the report needs clear organization before publishing.  Thank you for expressing this need. As no details have been provided regarding what the reviewer thinks clear organization of this paper would mean, it is hoped that the reorganization that has been provided relates to what the reviewer expects.

This article discussed Prof Rowe’s research program and self-directed research method.  It is not clear which is the focal point of the paper. Rowe’s research program was one of the ways that he supported self-direction in his graduate students by providing means of applying group theory that make the type of research graduate students themselves could engage in more powerful and permit them to come to significant results sooner. The focal point is self-directed learning—Rowe’s research program was the major part of what he provided to graduate students in his promoting of self-directed learning. I hope this is now clear from how the paper has been reorganized.

In addition, the description of Self-Directed Learning needs more explanation.  This has now been provided in lines 57-208. There should be specific examples of this philosophy of education that differs from the perspective of a particular graduate student.  Thank you for indicating that there need to be specific examples. There are a number of additional research papers that have been referenced regarding Self-Directed Learning in new section 2 (see references 24-36).

The paper would be much stronger if there were a detailed discussion on the traditional way to mentor a student compared with Self-Directed Learning.  Several such examples would make the paper much more substantial. Thank you for this point. This detailed discussion has been added in lines 105-110, including two new references (24 and 25).

Round 2

Reviewer 3 Report

This paper seems to be composed of two different topics.  The article begins by discussing self-directed learning.  The second section of the report discusses Prof J. Rowe.  The two areas do not seem to be sufficiently merged.  I think the authors would be better at choosing only one of the topics.  The transition between the subjects is too disjointed.

I have two separate comments:

The education of a graduate experimentalist is quite different.  Often these students are in large groups.  This situation is not discussed.  Teamwork is one of the most critical skills in this subfield.  

Also, Prof. Rowe's name is not mentioned until the middle of the abstract.  His name is not mentioned until line 223.  In addition, many initial sections concentrate on his research and not his teaching methods.

There are just too many disjoint topics for me.

Author Response

The following is the response to reviewer 3. Each of the reviewer’s concerns will be listed in plain text and the author’s response will follow in bold.

This paper seems to be composed of two different topics.  

Thank you for your comment that this paper seems to be composed of two different topics. The topic of the paper is self-directed learning for STEM. The commitment David. J. Rowe had to self-directed learning as a process in physics graduate education is an example of that how self-directed learning can be encouraged in physics graduate education. It is intended and presented as an example, not a separate topic.

The article begins by discussing self-directed learning.  

It is correct that the paper begins by discussing self-directed learning. This is because self-directed learning is the focus of the submission.

The second section of the report discusses Prof J. Rowe.

It is correct that the second section discusses Prof. David J. Rowe. It does so presenting the process he used for his own learning and in mentoring his students as a particularly appropriate and successful example of self-directed learning in STEM, not as a second topic. 

The two areas do not seem to be sufficiently merged.  

Thank you for your comment. If the paper is read as a discussion of self-directed learning, on the one hand, and of Prof. David J. Rowe, on the other, it is understandable that these two areas would not be seen as sufficiently merged. However, the self-direction Prof. Rowe himself demonstrated in his own research program and his manner of supporting self-directed learning in his graduate students is intended and presented as an example of self-directed learning in physics graduate education, not as a separate area.

I think the authors would be better at choosing only one of the topics.  The transition between the subjects is too disjointed.

Thank you for your comment that, if this paper is read as presenting two topics, only one of the topics should be chosen as the topics are found disjointed. However, there is only one topic of the paper—self-directed learning. The self-directed learning process of Prof. Rowe in his own research program and with his graduate students is one particularly successful example of how to support self-directed learning in physics graduate education. It is as an example that there is good reason for the part on Prof. David Rowe to remain in a discussion of self-directed learning in STEM.

I have two separate comments:

Thank you for your two separate comments; they will be responded to individually to follow.

The education of a graduate experimentalist is quite different.  Often these students are in large groups.  This situation is not discussed.  Teamwork is one of the most critical skills in this subfield.

Thank you for your comment that the education of graduate experimentalists is quite different. To meet this concern, the following has been added to lines 1175-1182: “As well, they generally conduct their research in large teams. There have been few studies on self-directed learning in large research teams. However, what has been done indicates that the larger the research group, the less graduate students self-direct their learning leading to frustration on the part of these graduate students as they believe it is difficult for them to fully participate in their learning. What has also been noted is that supervisors of these large teams are generally unaware that their graduate students are frustrated and experience their lack of self-direction negatively [161].”

The reference that goes along with citation 161 is as follows.

  1. Lohman, M.C.; Finkelstein, M. Designing groups in problem-based learning to promote problem-solving skill and self-directedness. Instruct. Sci.2000, 28, 291–307. https://doi.org/10.1023/A:1003927228005

As well, this sentence was added to lines 1197-1199: “One way in which these could be incorporated is through virtual laboratory technologies that have been found effective [162].

The reference that goes along with citation 162 is as follows.

  1. Trúchly, P.; Medvecký, M.; Podhradský, P.; El Mawas, N. STEM education supported by virtual laboratory incorporated in self-directed learning process. J. Electric. Eng. 2019, 70, 332-44. https://doi.org/10.2478/jee-2019-0065

Also, Prof. Rowe's name is not mentioned until the middle of the abstract.  

To account for the reviewer’s concern that Prof. Rowe’s was not mentioned by name in the abstract until half-way, in line 10, “One” was changed to “David J. Rowe, a”.

His name is not mentioned until line 223.  

To meet the reviewer’s concern that Prof. Rowe’s name is not mentioned until line 223, line 57 changed “one scientist” to “Prof. David J. Rowe—a theoretical nuclear physicist—”.

In addition, many initial sections concentrate on his research and not his teaching methods.

Thank you for your point that many initial sections concentrate on Prof. Rowe’s research and not his teaching methods. As pointed out in the text, Prof. Rowe was an excellent example of a self-directed learner himself. His research program demonstrates how he learned as such a learner. To make this point more clearly, two changes were made. The first is to line 58, where “to encourage”  was changed to “with respect to his own research program and to mentor”. The second is to lines 60-62 where the following point was added: “This will be accomplished by initially examining self-directed learning in general and then looking specifically at the type of self-directed learning process supported by Prof. Rowe.”

There are just too many disjoint topics for me.

Thank you for your comment that there are too many disjointed topics. There are a number of topics discussed. However, given that the argument presented in the paper is that self-directed learning is essential to successful graduate education and that Professor David J. Rowe developed an exemplary process of self-directed learning for himself and as a mentor to graduate students, the number of topics discussed is considered necessary.